# Impact of the implementation of Identification-Situation-Background-Assessment-Recommendation (ISBAR) tool to improve quality and safety measure in a lithotripsy and endourological unit

**Patricia Gadea-Company**[1,2☯]*, **Carmen Casal Angulo**[1,3☯], **Clara Hurtado Navarro**[1,4☯]

**1** Faculty of Nursing and Podiatry, University of Valencia, Valencia, Spain, **2** Department of Lithotripsy and Endourology Unit, La Fe University and Polytechnic Hospital, Valencia, Spain, **3** Department of Sanitary Emergencies in Comunidad Valenciana SES-CV, Valencia, Spain, **4** Department of Teaching, University Hospital Doctor Peset, Valencia, Spain

☯ These authors contributed equally to this work.

\* patri.gadea@gmail.com

## Abstract

### Introduction

A lack of professional communication and collaboration may be one of the main causes of medication errors. The objective was to evaluate the results of the implementation of ISBAR as a communication and safety tool in a Lithotripsy and Endourologic Unit of a tertiary public hospital.

### Methods

A total of 457 patients were included in a retrospective study from 2014 to 2019. Patients were divided into two groups: group A (357 patients) in which an endourological procedure was performed before march of 2018 (without the implementation of ISBAR tool) and Group B (100 patients) with the implementation of ISBAR tool. The inclusion criteria were patients accepted for surgical intervention by anaesthesiology Department and operated in the period of the study. The variables analysed included number of procedures, global, intraoperative and postoperative complications rate, urinary infection or sepsis, NPR (FMEA), percentage of suspended surgical patients and hospital stay.

### Results

The postoperative complications showed no significant differences between groups, but a trend to diminishing was seen in the complication in the group B. The sepsis reduced its incidence and it was close to significant difference. The operative time was shorter in group B 119,11min (114,63–123,59) vs 115,11min (109,63–121,67) p = 0,3. The reduction in the main postoperative complication (sepsis) explained the lower hospital stay for group B. The severe adverse events detected were reduced completely.

**Data Availability Statement:** All relevant data are within the paper and its Supporting Information files.

**Funding:** The authors received no specific funding for this work.

**Competing interests:** The authors have declared that no competing interests exist.

## Conclusions

ISBAR tool was an effective patient safety tool improving quality care. To provide safe patient care and improving quality is indispensable an effective communication flow.

## Introduction

Teamwork and communication (collaborative practice) between healthcare personnel is essential for achieving optimal patient outcomes [1]. Patient safety is defined as defense against unnecessary injury resulting from the health service's efforts or absence of efforts.

However, unclear and imprecise communication is common among healthcare personnel [2]. A lack of professional communication and collaboration may be one of the main causes of medication errors [3]. Poor communication is a contributing factor in more than 60% of all hospital adverse events as the Joint Commission announced [4].

The mnemonic Identification, Situation, Background, Assessment, Recommendation (ISBAR) was established to improved safety in transfer of patient information. ISBAR is the acronym of Identification, Situation, Background, Assessment, Recommendation. It is a patient safety communication tool that facilitate the communication between healthcare personnel.

It was first introduced in acute care and after expanded to other teams. ISBAR communication tool have shown to many authors that to use it develops the teamwork, and patient safety. It provides more scope for profesional discussions to make a plan together on treatment and further plans for the patient [4]. ISBAR communication tool strengthens dialogue between team members. It promotes the improving of patient satisfaction and outcomes making common decisions and conflicts resolution between team members [5, 6].

Qualitative findings identified four main themes that explicate the utility of ISBAR as an interprofessional communication tool across clinical and non-clinical settings: common language between different professions, efficient organisation of the information, facilitation of collaborative team-based communication (including shared decision making and conflict resolution) and finally, versatility (permitting use in different formats such as face to face discussions, group presentations, email communication and writing of approval papers) [7]. This communication tool is a reliable and validated. It has shown a decrease in adverse events in a hospital setting, enhancemend in promotion of patient safety and communication within health care providers [4]. SBAR (Situation, Background, Assessment, Recommendation) has been recognize as an effective communication tool for patients by the Joint Commission, Agency for Healthcare Research and Quality (AHRQ), Institute for Health Care Improvement (IHI), and World Health Organization (WHO).

The ISBAR tool was implemented in our department in order to increase the safety of the patient in the operating room, in which we perform mainly endourological procedures. For that, our main aim in this paper was to evaluate the results of the implementation of ISBAR as a communication and safety tool in a Lithotripsy and Endourologic Unit of a tertiary Spanish public hospital.

## Material and methods

A total of 457 patients were included in a retrospective study with a prospective collecting data from January of 2014 to December of 2019 in a tertiary reference Lithotripsy and Endourological Unit. The prospective study means that we will keep including ISBAR tool in our unit to

study the communication tool effectiveness. Patients were divided into two groups: group A (357 patients) in which an endourological procedure was performed before march of 2018 (without the implementation of ISBAR tool) and Group B (100 patients) with the implementation of ISBAR tool, from march of 2018 to December of 2019.

The inclusion criteria were patients accepted for surgical intervention by anaesthesiology Department and operated in the period of the study. The endourological procedures included was the percutaneous nephrolithotomy (PCNL), because was the procedure with higher probability of surgical and postoperative complications. PCNL has been considered by many authors as the first line of treatment in renal stones [8]. The patient position normally is prone position and it has been studying that it the safest access from the skin to the renal collecting system. In this kind of surgeries Biplanar fluoroscopy with the use of the C rotational fluoroscopy–arm will be use for renal puncture. The dilation of the nephrostomy tract is the most important element in this kind of surgeries. It will be use as normal Alken dilators or Amplatz dilators after that, the surgeon will ride intracorporal lithotrites (flexible and rigid instruments) to remove the stones which is the main objective of the surgery. After removing the stones a nephrostomy tube will be place if it is necessary to make sure that the kidney drains [9]. It has been discussed for many authors about anesthesics technique in this kind of procedures. After the induction of general anesthesia has shown less neurological complications [10]. According to some authors, it seems that using general anesthesia vs spinal reduces the requirement for analgesia for patients in postoperative periods [11].

The exclusion criteria were endourological procedures with low probability of complications (ureteroscopy and retrograde intrarenal surgery) and patients with high surgical risk due to their comorbidity (to avoid the bias of the severe medical conditions could have in the complication rate). High surgical risk was considered in patients with urinary diversion, spine injuries, uncorrected coagulation disorders and severe lung and heart dysfunction.

The lithiasis and patient characteristics were compared between groups. The main variables analysed included number of procedures, global complications rate, intraoperative and postoperative complications, Clavien classification of complications, percentage of fever, urinary infection or sepsis, NPR (modal analysis of failures and effects), percentage of suspended surgical patients in the operating room and hospital stay.

The ISBAR tool was adapted to the Lithotripsy and Endourological Unit following the ISBAR tool structure by PG and AB. A specific template was designed (Fig 1) and approved in the management session of the Unit. The nursing office prepares every day the ISBAR of the following day. The ISBAR is reported five minutes before starting the surgical session. All the participants in the operating room are cited, including anaesthesiologist, surgeons, nurses, auxiliary nursing and stretcher-bearer. Each step of ISBAR is resumed in the session; *Identification* includes the name, gender and age. *Situation* evaluates special conditions of the patient (disabilities, etc), *Background* analyses the medical history of the patient, *Assessment* includes the diagnosis including the affect side, planned treatment and the material check list for each surgery and, finally *Recommendations* summarizes the specific treatments and the antibiotic prophylaxis according to the results of previous cultures.

A modal analysis of failures and effects was developed in the Unit office for surgical procedures in 2017, as a proactive risk management. Potential risks were identified in the office and discussed between all the staff in the management sessions. Specific measures were adopted by each identified risk and, 6 months later, the risks were evaluated again. The NPR was used as variable in order to calculate the reduction of the risk between groups. NPR is the product of severity, probability of frequency and capacity of detection. A scale from 1 to 5 was used for each factor. The NPR was calculated in the initial phase (risk identification) and after of implantation of the specific measures.

| ISBAR | |
|---|---|
| **Identification** | Name<br>Age<br>Sex<br>Location |
| **Situation** | Current problem and procedure to be carried out.<br>urgent / non-urgent.<br>If urgent: reason. |
| **Background** | Allergy history<br>Urine Culture<br>Analytics (hemostasis, creatinine)<br>Personal history (Hypertension, Diabetes, ...)<br>Treatments |
| **Assesment** | Special considerations<br>(example: if there is something special in the<br>procedure to be performed) |
| **Recommendation** | Surgical specific recommendations<br>(eg notify Dr. to see and / or operate(specific<br>material) |

**Fig 1. ISBAR.** Structured communication in a surgical area.

For statistical analysis, the categorical variables were resumed as frequency and percentage. The continuous variables were resumed using the average and 95% confidence interval (CI 95%). A $X^2$ test was used for comparing qualitative variables and a T-Student test for continuous variables. A statistical significance of 0,05 was considered ($p < 0,05$).

## Ethical aspects

The data of the participants was treated in accordance with current data protection legislation. He secured the anonymity of the participants and the confidentiality of their data. They signed the informed consent for the surgery where they were informed verbally of the objectives of the study. It was collect in our data as the informed consent from the surgery and anesthetics. With it, they expressed their desire to participate in it. In carrying out the study, we have followed the ethical principles of the Declaration of Helsinki [12]. The Ethic Committee approval was considered: Comite Ético de Investigación Biomédica del Hospital Universitario y Poltécnico La Fe with registered number 220-281-1 and date 08 May 2020.

## Results

Both groups were comparable for gender, age, long X-ray stone diameter and number of lithiasis treated. Only type 1 diabetes was more frequent in group A (Table 1).

There was no difference between groups in the global complications rate. A total of 109 patients (31,1%) in group A had complications and 31 patients (31%) in group B (p = 0,9). There were also no differences in the type of complications (intra o postoperative) between groups (p = 0,4). The severity of complications (Clavien < 3) was similar between groups (22,7% vs 19% p = 0,4). The individualized analysis of intraoperative complications is shown in the Table 2.

The postoperative complications showed no significant differences between groups, but a trend to diminishing was seen in the complication in the group B. In spite of that, the sepsis reduced its incidence and it was close to significant difference. The low number of

**Table 1. Comparative analysis of the main patient and stone characteristics.**

| Variables | Group A (no ISBAR) n (%) | Group B (ISBAR) n (%) | p |
|---|---|---|---|
| Gender (Female) | 173 (50,7) | 46 (46%) | 0,4 |
| Type 1 diabetes | 14 (4) | 9 (9) | 0,03 |
| Type 2 diabetes | 38 (10,8) | 5 (5) | 0,03 |
|  | **Group A (no ISBAR) x (CI 95)** | **Group B (ISBAR) x (CI 95)** |  |
| Age | 51,93 (49,74–54.12) | 57,60 (53,8–61,3) | 0,4 |
| BMI | 27,7 (26,76–28,6) | 26,54 (25,22–27,85) | 0,82 |
| Stone size | 27,43 (25,17–29,69) | 23,14 (20,5–25,78) | 0,9 |
| Hounsfield Units | 1000,81 (953,68–1047,95) | 1063,36 (981,98–1144,75) | 0,4 |
| Number of stones | 1,95 (1,66–2,25) | 2,14 (1,73–2,54) | 0,8 |

**Table 2. Individualized analysis of intraoperative complications.**

| Variables | Group A (no ISBAR) n (%) | Group B (ISBAR) n (%) | p |
|---|---|---|---|
| No complications | 302 (86) | 88 (88) | 0,5 |
| Bleeding | 33 (9,4) | 9 (9) | 0,5 |
| Collecting system perforation | 9 (2,6) | 2 (2) | 0,5 |

The operative time was shorter in the group B [119,11min (114,63–123,59) vs 115,11min (109,63–121,67) p = 0,3] but without statistical differences.

**Table 3. Comparative analysis of postoperative complications between groups.**

| Variables | Group A (no ISBAR) n (%) | Group B (ISBAR) n (%) | p |
|---|---|---|---|
| Sepsis | 26 (7,4) | 3 (3) | 0,08 |
| Hematuria | 16 (4,6) | 4 (4) | 0,5 |
| Perirenal bleeding | 3 (0,9) | 0 (0) | 0,5 |
| Fistula | 1 (0,3) | 0 (0) | 0,5 |
| Pneumonia | 1 (0,3) | 0 (0) | 0,5 |
| Transfusion | 11 (3,1) | 2 (2) | 0,4 |
| Embolization | 7 (2) | 1 (1) | 0,44 |

**Table 4. Percentage of suspended patients in operating room over time (from 2018).**

|  | % suspended surgical patients |
|---|---|
| 2018 | 2,91 |
| 2019 | 2,49 |
| 2020 (until April 2020) | 2,06 |

complications was the main cause of not reaching the statistical significance. The comparative analysis of postoperative complications between groups is shown in the Table 3.

The reduction in the main postoperative complication (sepsis) explained the lower hospital stay for group B. The hospital stay was of 2,97 days (2,66–3,27) and 2,05 days (1,66–2,44) for group A and B, respectively (p = 0,03). The percentage of suspended patients was decreasing progressivity. The main results are shown in the Table 4.

The modal analysis of failures and effects identified several risks in the PCNL process: incorrect identification of the affected side and wrong antibiotic prophylaxis. The initial Risk

**Table 5. Modal analysis of failures and effects.**

| Failure mode | Effect | S | F | D | Initial NPR | Recommended action | S | F | D | Final NPR |
|---|---|---|---|---|---|---|---|---|---|---|
| Wrong side | Wrong treatment | 5 | 3 | 3 | 45 | ISBAR and surgical checklist | 5 | 1 | 1 | 5 |
| Wrong prophylaxis | Postoperative Sepsis | 4 | 2 | 3 | 24 | ISBAR | 4 | 1 | 1 | 4 |

NPR (Risk Priority Number), S (severity), F (frequency), D (detection). NPR = S x F x D

Priority Number (NPR), the corrective measures and the final NPR is shown in the Table 5. The corrective measures reduced the frequency and increased the detection capacity.

The severe adverse events detected were reduced completely (2 severe adverse effects before ISBAR implantation and 0 severe adverse effects after implantation).

## Discussion

In our knowledge, it is the first article that has evaluated the impact of the implantation of an ISBAR tool in a Lithotripsy and Endourological Unit. The implementation of the ISBAR tool was done in the context of a continuous improvement project based on Lean methodology.

The implementation of the ISBAR tool communication arises from the idea of increasing the patient and surgical procedures safety. This initiative was focused on patient safety because it was observed that "some" adverse events occurred because of poverty communication of the team. In the same way, the information to the patient is mandatory in healthcare, especially in a surgical department. The additional information provided previously in nursing office about the technique, process and the attention afford an added value [13, 14]. Using ISBAR increases the awareness of users' own structured communication and expertise and allows them to obtain a quicker overview of patient situations [4].

The application of SBAR tool communication in our department showed that the operative time was shorter. There was a clear trend to reduce the operating time in the group, because the preparation of the patient and antibiotic prophylaxis was planned in the ISBAR session. It reduced the time consumed in the initial phase of the surgical procedure.

The postoperative complications showed no significant differences between groups, but a trend to diminishing was seen in the complication in the group B. The individualized analysis of the complications showed that the sepsis reduced its incidence and it was close to significant difference. Perhaps, the low incidence of sepsis in both groups was the main cause of not reaching the statistical significance. In our opinion, the use of ISBAR tool allowed to choose and discuss the properly antibiotic prophylaxis between the surgical team, improving the communication and reducing the probability of failure in its administration.

The reduction of postoperative sepsis explained the lower hospital stay. The sepsis is the most severe complication in endourological procedures, and it can suppose long hospitalization stays including the intensive care. The reduction of its incidence supposes an important improvement of the patient safety and a reduction of the procedure costs. In spite of the sepsis depends on many factors, to get a sterile urine at the moment of the surgery is one of the most important aspects in the prevention of the development of this complication [15, 16].

The use of ISBAR tool obtained a progressive reduction in the percentage of suspended patients. Detecting problems during the SBAR allowed to make corrective decisions that avoid the suspension of patients in the operating room. For example, an incomplete correction of blood disorders (anticoagulants) can let us to change in the order of the surgical planning and adopt the necessary measures for its proper correction avoiding the immediate suspension.

In other hand, the ISBAR was the main tool used in the modal analysis of failures and effects that allowed to reduce the NPR in endourological procedures. The modal analysis of

failures and effects is a risk proactive management that identify the safety risk before they occur. The identification of security gaps in surgical process reduce significantly the adverse events. In our experience, the implementation of ISBAR tool was an effective measure to reduce the NPR in endourological surgeries. The initial risks identified (wrong side or wrong antibiotic prophylaxis) had a high NPR score, one with an extreme risk (score of 45) and the other one with a severe risk (score of 24). The implementation of ISBAR achieved a reduction in NPR in both risks (score less than 10), classifying them after the intervention as an acceptable risk. So, ISBAR was the main measure that let to reduce the severity of the risks previously identified. The adverse events observed were reduced significantly in our Unit after applying ISBAR. This reduction was observed as well by Randmaa M et al, who decreased the adverse events in more than 65% in an anaesthesiology department after implementing SBAR tool [17].

Muller M et al, in a systematic review demonstrated the effectiveness of SBAR tool on patient outcome, although he recognized in his review that the evidence is limited [18]. Our study adds more evidence about the ISBAR tool usefulness on patient outcome, mainly in surgical safety.

The main inconvenience were to adopt the communication tool and sustain structured communication because it was necessary a culture change by all health care providers. They thought using ISBAR was time-consuming considering that this communication structure was not incorporated into work routine in the unit.

However, the implementation of ISBAR improved job satisfaction and safety climate. It was observed as well by Ting WH et al in an obstetric department [19]. It come out that it was difficult to go behind the structure automatically. After several ISBAR simulations they found it easier, therefore we can estímate that training was necessary to incorporate ISBAR as a routine.

Consequently, the support and acceptance of health personnel was one of the biggest trouble and it was required alignment between the profesional team and all categories. As Sahid S et al comment, the use of ISBAR communication tool requires educational training and culture change to sustain its clinical use [20].

The main strengths of our study were the possibility of objectively measuring the usefulness of implementing a continuous improvement tool such as the ISBAR with different clinical variables. It allowed us to demonstrate that ISBAR can increase the patient safety and improve the quality of care only with a structured transmission of the information in a short period of time before surgical session. Therefore, it can be generalized and applied to other surgical specialties. The main limitations of our study were the low severe complications rate (sepsis) observed, that made it difficult to obtain significant differences. In spite of that, the most severe complication showed a trend very closed to the statistical significance. Therefore, the reduction of percentage of suspended patients and the hospital stay supported the effectiveness of the ISBAR tool in our study [21].

## Conclusions

So, according to our results, ISBAR tool was an effective patient safety tool and it can be applied in nursing assessment and healthcare. Providing safe patient and improving quality care is given by an effective communication flow. In fact, the adverse events related to communication errors can be reduced and brings the opportunity to make our work better making better decisions for the patient treatments. For its implementation and incorporation, it requires an adaptation period and a culture change in our day to day.

## Acknowledgments

All authors listed on the manuscript have contributed sufficiently for the project to be included as authors.

Patricia Gadea-Company

gacompa@uv.es

Faculty of Nursing and Podiatry. University of Valencia (Spain).

Department of Lithotripsy and Endourology Unit. La Fe University and Polytechnic Hospital. Valencia (Spain).

Carmen Casal Angulo

m.carmen.casal@uv.es

Faculty of Nursing and Podiatry. University of Valencia (Spain). Department of sanitary emergencies in Comunidad Valenciana SES-CV. Valencia (Spain).

Clara Hurtado Navarro

clara.hurtado@uv.es

Faculty of Nursing and Podiatry. University of Valencia (Spain).

Department of teaching. University Hospital Doctor Peset. Valencia (Spain).

## Author Contributions

**Conceptualization:** Patricia Gadea-Company.

**Data curation:** Patricia Gadea-Company.

**Formal analysis:** Patricia Gadea-Company.

**Funding acquisition:** Patricia Gadea-Company.

**Investigation:** Patricia Gadea-Company.

**Methodology:** Patricia Gadea-Company.

**Project administration:** Patricia Gadea-Company.

**Resources:** Patricia Gadea-Company.

**Software:** Patricia Gadea-Company.

**Supervision:** Patricia Gadea-Company, Carmen Casal Angulo, Clara Hurtado Navarro.

**Validation:** Patricia Gadea-Company, Carmen Casal Angulo, Clara Hurtado Navarro.

**Visualization:** Patricia Gadea-Company, Clara Hurtado Navarro.

**Writing – original draft:** Patricia Gadea-Company.

**Writing – review & editing:** Patricia Gadea-Company, Carmen Casal Angulo.

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
