## [Decision Letter · Decision Letter 0]

8 May 2023

PONE-D-22-33805We are submitting the manuscript entitled “Impact of the implementation of Identification-Situation-Background-Assesment-Recommendation (ISBAR) tool as improving quality and safety measure in a lithotripsy and endourological Unit”.PLOS ONE

Dear Dr. GADEA-COMPANY,

Thank you for submitting your manuscript to PLOS ONE. After careful consideration, we feel that it has merit but does not fully meet PLOS ONE’s publication criteria as it currently stands. Therefore, we invite you to submit a revised version of the manuscript that addresses the points raised during the review process.

We look forward to receiving your revised manuscript.

Kind regards,

Antonino Maniaci

Academic Editor

PLOS ONE

Journal Requirements:

2. In your Methods section, you describe your study as “retrospective study with a prospective collecting data”. Please clarify what you mean by prospective collecting data and add the according details in your revised manuscript."

3. Please ensure that you have specified (1) whether consent was informed and (2) what type you obtained (for instance, written or verbal, and if verbal, how it was documented and witnessed). If your study included minors, state whether you obtained consent from parents or guardians. If the need for consent was waived by the ethics committee, please include this information.

5. Please include a caption for figure 1.

Additional Editor Comments (if provided):

interesting paper, perform the revisions required .

Reviewers' comments:

Reviewer's Responses to Questions

**Comments to the Author**

1. Is the manuscript technically sound, and do the data support the conclusions?

Reviewer #1: Yes

Reviewer #2: No

Reviewer #3: Partly

2. Has the statistical analysis been performed appropriately and rigorously? 

Reviewer #1: Yes

Reviewer #2: No

Reviewer #3: Yes

3. Have the authors made all data underlying the findings in their manuscript fully available?

Reviewer #1: Yes

Reviewer #2: Yes

Reviewer #3: Yes

4. Is the manuscript presented in an intelligible fashion and written in standard English?

Reviewer #1: Yes

Reviewer #2: No

Reviewer #3: Yes

5. Review Comments to the Author

Reviewer #1: I read with great interest the manuscript by Gadea-Company et al. on the implementation of ISBAR tool to improve quality and safety measures. The paper is relevant and sound. However, I have some minor issues to be addressed.

Title. Please modify into "IMPACT OF THE IMPLEMENTATION OF IDENTIFICATION-SITUATION-BACKGROUND-ASSESSMENT-RECOMMENDATION (ISBAR) TOOL TO IMPROVE QUALITY AND SAFETY MEASURE IN A LITHOTRIPSY AND

ENDOUROLOGICAL UNIT"

Introduction.

- Acronyms should be written in full the first time they appear in text. Please modify into: "The mnemonic tool Identification-Situation-Background-Assessment-Recommendation (ISBAR) was established to improve..."

- You should mention the surgical technique in use for percutaneous nephrolithotomy (doi: 10.1007/s00345-017-2001-0) and state that it could be performed under general (doi: 10.4111/kju.2013.54.3.172) and spinal anesthesia (doi: 10.23736/S0375-9393.22.16969-5). Please briefly discuss and add these 3 references.

Results.

- Please specify the acronym NPR in Table 5.

Discussion.

- Simulation for surgical training is also a valid and well known tool to increase patient safety and quality of surgical procedures (doi: 10.1016/j.nec.2014.11.002 - doi: 10.1136/qshc.2010.042424). Please discuss.

Reviewer #2: Aim of this manuscript is to assess the impact of introduction of surgical safety checklist on perioperative outcomes of endourology procedures in a large referral center. Topic is relevant and efforts of Authors are praiseworthy; however, the manuscript shows several inherent flaws.

-Firstly, the retrospective nature of the study impacts on the overall reliability of results. Moreover, the Authors only included patients treated with "high-risk" endoscopic procedures (PCNL) due to the increase likelihood of periop complications and also they excluded patients with high surgical risks. These arbitrary selections negatively impact on the reliability of results.

-The Authors concluded that the introduction of surgical safety checklist reduced rate of sepsis and that also influenced lenght of hospital stay. However, difference in terms of sepsis rates between the two study groups was not statistically significant. Therefore, these conclusions cannot be drawn.

-Preoperative and postoperative features should be reported as median (IQR) rather than mean (SD).

Reviewer #3: 1-Why do you compared two non-equal numbers of patients?

2-As long as it is in urological unite why should not include all the types od surgery, some time easy cases have miss-conduction.

3-Bleeding was included in the complication!

4-There are some minor differences in your outcomes, but they are non-significant enough to support your conclusion.

5-How you reached the conclusion: effective safety tool.

6-Who secure the anonymity of the participants?

7-How both groups are comparable, and data shows female 173 Group A and 46 Group B, meanwhile the 173 is not 50.7% of 357 patients.

8-Both DM 1 and 2 are more frequent and significant. Do you think this may affect the outcomes?

9-Sepsis has many predisposing factors: Do you think it has been reduced by ISBAR only?

10- Postoperative hospital stay will be prolonged also due to leakage and bleeding not only sepsis.

11- Reference 7 and 17need correction.

6. PLOS authors have the option to publish the peer review history of their article (what does this mean?). If published, this will include your full peer review and any attached files.

Reviewer #1: No

Reviewer #2: No

Reviewer #3: No

---

## [Author Response · Author response to Decision Letter 0]

17 May 2023

Dear editor,

First, we would like to thank you once again for the time you have taken to review our manuscript entitled “Impact of the implementation of Identification-Situation-Background-Assesment-Recommendation (ISBAR) tool to improve quality and safety measure in a lithotripsy and endourological Unit”. PONE-D-22-33805

We have made a number of changes to the manuscript, according to the editor’s and reviewers’ comments and suggestions. We have now provided a detailed list of these changes. In the manuscript, we have made different changes, which are marked in yellow. Your comments were very useful and have allowed us to improve our work to make it more interesting, attractive, and useful for the scientific community.

Finally, we would like to thank you once again for your work, as your contributions have allowed us to improve this work and thus improve existing knowledge in the analysis of quality in initial pediatric trauma care. We hope that the revised manuscript will be accepted for publication in the Nursing in Critical Care Journal.

Sincerely,

The authors

---

## [Editor Report · Decision Letter 1]

19 May 2023

Impact of the implementation of Identification-Situation-Background-Assesment-Recommendation (ISBAR) tool to improve quality and safety measure in a lithotripsy and endourological Unit.

PONE-D-22-33805R1

Dear Dr. GADEA-COMPANY,

We’re pleased to inform you that your manuscript has been judged scientifically suitable for publication and will be formally accepted for publication once it meets all outstanding technical requirements.

Kind regards,

Antonino Maniaci

Academic Editor

PLOS ONE
---

## [Editor Report · Acceptance letter]

23 May 2023

PONE-D-22-33805R1 

Impact of the implementation of Identification-Situation-Background-Assessment-Recommendation (ISBAR) tool to improve quality and safety measure in a lithotripsy and endourological unit. 

Dear Dr. Gadea-Company:

I'm pleased to inform you that your manuscript has been deemed suitable for publication in PLOS ONE. Congratulations! Your manuscript is now with our production department. 

Kind regards, 

on behalf of

Dr. Antonino Maniaci 

Academic Editor

PLOS ONE